# Structural Characterization of a Minimal Antibody against Human APOBEC3B

**DOI:** 10.3390/v13040663

**Published:** 2021-04-12

**Authors:** Heng Tang, Özlem Demir, Fredy Kurniawan, William L. Brown, Ke Shi, Nicholas H. Moeller, Michael A. Carpenter, Christopher Belica, Kayo Orellana, Guocheng Du, Aaron M. LeBeau, Rommie E. Amaro, Reuben S. Harris, Hideki Aihara

**Affiliations:** 1Department of Biochemistry, Molecular Biology, and Biophysics, University of Minnesota, Minneapolis, MN 55455, USA; merlinczar@gmail.com (H.T.); kurni012@umn.edu (F.K.); brown344@umn.edu (W.L.B.); shixx023@umn.edu (K.S.); nicholas.h.moeller@gmail.com (N.H.M.); mcarpent@umn.edu (M.A.C.); belic028@umn.edu (C.B.); orellk@umn.edu (K.O.); 2Institute for Molecular Virology, University of Minnesota, Minneapolis, MN 55455, USA; 3Masonic Cancer Center, University of Minnesota, Minneapolis, MN 55455, USA; 4Key Laboratory of Industrial Biotechnology, Ministry of Education, School of Biotechnology, Jiangnan University, 1800 Lihu Road, Wuxi 214122, China; gcdu@jiangnan.edu.cn; 5School of Biotechnology, Jiangnan University, 1800 Lihu Road, Wuxi 214122, China; 6Department of Chemistry and Biochemistry, University of California, San Diego, La Jolla, CA 92093, USA; ozlemd2021@gmail.com (Ö.D.); ramaro@ucsd.edu (R.E.A.); 7Howard Hughes Medical Institute, University of Minnesota, Minneapolis, MN 55455, USA; 8The Key Laboratory of Carbohydrate Chemistry and Biotechnology, Ministry of Education, Jiangnan University, 1800 Lihu Road, Wuxi 214122, China; 9Department of Pharmacology, University of Minnesota Medical School, Minneapolis, MN 55455, USA; alebeau@umn.edu

**Keywords:** APOBEC3B, antiviral innate immunity, cancer mutagenesis, crystal structure, DNA cytosine deaminase, molecular dynamics simulation, monoclonal antibody, protein-protein docking, scFv, tumor evolution

## Abstract

APOBEC3B (A3B) is one of seven human APOBEC3 DNA cytosine deaminases that restrict viral infections as part of the overall innate immune response, but it also plays a major role in tumor evolution by mutating genomic DNA. Given the importance of A3B as a restriction factor of viral infections and as a driver of multiple human cancers, selective antibodies against A3B are highly desirable for its specific detection in various research and possibly diagnostic applications. Here, we describe a high-affinity minimal antibody, designated 5G7, obtained via a phage display screening against the C-terminal catalytic domain (ctd) of A3B. 5G7 also binds APOBEC3A that is highly homologous to A3Bctd but does not bind the catalytic domain of APOBEC3G, another Z1-type deaminase domain. The crystal structure of 5G7 shows a canonical arrangement of the heavy and light chain variable domains, with their complementarity-determining region (CDR) loops lining an antigen-binding cleft that accommodates a pair of α-helices. To understand the mechanism of A3Bctd recognition by 5G7, we used the crystal structures of A3Bctd and 5G7 as templates and computationally predicted the A3B-5G7 complex structure. Stable binding poses obtained by the simulation were further tested by site-directed mutagenesis and in vitro binding analyses. These studies mapped the epitope for 5G7 to a portion of C-terminal α6 helix of A3Bctd, with Arg374 playing an essential role. The same region of A3Bctd was used previously as a peptide antigen for generating a rabbit monoclonal antibody (mAb 5210-87-13), suggesting that this region is particularly immunogenic and that these antibodies from very different origins may share similar binding modes. Our studies provide a platform for the development of selective antibodies against A3B and other APOBEC3 family enzymes.

## 1. Introduction

The APOBEC3 (A3) family of single-stranded (ss) DNA cytosine deaminases comprises seven enzymes in most humans that have vital innate immune functions in restricting the replication of various DNA-based viruses and transposable elements [1,2]. The A3 enzymes consist of either a single or tandem repeat of a conserved zinc-dependent cytosine deaminase fold, with each member exhibiting clear target DNA sequence preferences and distinct subcellular localization patterns [2,3]. However, despite beneficial antiviral functions, several A3 enzymes have been implicated in cancer genome mutagenesis and tumor evolution [3,4,5]. In particular, several studies strongly indicate that the nuclear-localized family member APOBEC3B (A3B) is a significant source of ongoing mutations in multiple human cancers, including those of the breast, head/neck, lung, bladder, and cervix [3,4,5]. Thus, studies on A3 expression, localization, and interaction partners are important to better understand both the mechanisms of host–virus interaction as well as those of cancer mutagenesis.

One of the challenges in studying A3 enzymes is the difficulty in obtaining specific antibodies against each unique family member. This is due to a limited number of immunogenic sequences and a high degree of homology between the different human A3 family members. Although a rabbit monoclonal antibody (mAb) against human A3B has been described and proven useful in immunohistochemistry and other applications, this reagent still cross-reacts with A3A and A3G due to epitope homology [6]. Thus, high-resolution structural information on A3B–antibody complexes would be highly valuable, as it has the potential to guide optimization to improve selectivity and ultimately result in highly specific and versatile antibodies. Here, we describe an A3B-binding, single-chain variable fragment (scFv) that was obtained through phage display screening, characterized in multiple immune assays, and resolved by X-ray crystallography. Furthermore, simulations of computationally predicted A3B-scFv complexes and site-directed mutagenesis combined to identify a key residue involved in the A3B-scFv interaction and suggest a plausible binding mechanism. Overall, these studies describe a novel reagent for further work on A3B and provide a platform for future engineering of highly specific antibodies.

## 2. Materials and Methods

### 2.1. Phage Display Screening for Anti-A3B scFvs

A scFv library derived from mouse spleen DNA [7] was screened against A3Bctd-QMΔloop3 [8] in the phage display format as described [9,10]. A3Bctd-QMΔloop3 protein was conjugated to biotin (1:2.5 ratio, EZ-Link NHS-PEG4-Biotin, ThermoFisher Scientific, Waltham, MA, USA). The biotinylated protein was bound to Streptavidin M-270 Dynabeads (ThermoFisher Scientific). A mouse scFv phage library (~2 × 10^9^ diversity) was enriched for A3B-binding by 4 rounds of selection with A3B-dynabeads. Each round of selection had an increased binding stringency with A3B ligand at 20, 10, 5, or 1 µg/mL. The beads were washed extensively with PBS/0.05% Tween-20. Captured phage were eluted with 100 mM ethanolamine, neutralized with 0.5 M Tris-HCl pH 7.4, and used to infect TG1 *E. coli.* Infected *E. coli* were collected and propagated with M13KO7 helper phage to re-amplify a sub-library enriched for binding to A3B binding. After the 4th round of enrichment, colonies were selected and grown in 96-well plates to express each scFv. A3B-binding scFv candidates were identified by ELISA. A3Bctd-QMΔloop3 (5 µg/mL) bound to Maxisorb plates (Nunc) was incubated with culture media from the selected colonies, followed by probing with anti-HA-Peroxidase (3F10, MilliporeSigma, Burlington, MA, USA). The color was developed using 1-Step Ultra TMB ELISA Substrate (ThermoFisher Scientific) and visualized at 450 nm. The 5G7 ScFv bound strongly in ELISA to A3Bctd-QMΔloop3 but not A3Gctd.

### 2.2. Protein Purification

The previously crystallized A3Bctd-QMΔloop3 and its Arg-to-Glu or Arg-to-Ala mutants were expressed with a C-terminal 6xHis tag in *E. coli* strain C41(DE3)pLysS and purified as described [8,11,12]. A3A-E72A was also produced as described [13]. A3Bctd-DM (L230K/F308K) and A3Gctd 2K3A [14] were expressed with a HRV 3C protease-cleavable GST-6xHis tag using pET-42(b) vector and purified using nickel-affinity chromatography and SEC. ScFv 5G7 was expressed using a pET-24a vector in SHuffle T7 (NEB) *E. coli* strain. The 5G7-clasp heavy and light chains were co-expressed using a modified pET-24a vector in *E. coli* strain BL21(DE3). The antibodies were purified as above, except in non-reducing conditions. Protein concentrations were determined based on UV absorption at 280 nm and calculated extinction coefficients from the amino acid sequences.

### 2.3. Protein Binding Studies (SEC, SPR, and BLI)

Various forms A3Bctd (based either on A3Bctd-QMΔloop3 or A3Bctd-DM, and with or without point mutations) and scFv 5G7 (with or without D62R) were co-injected into a 10/300 Superdex 75 or 200 column operating with a running buffer, 20 mM Tris-HCl and 0.5 M NaCl, at a flow rate of 0.4 mL/min at 4 °C. Protein co-elution was monitored by UV absorption at 280 nm and analyzing the collected fractions by SDS-PAGE. The SPR experiments were performed using Biacore 8K. ScFv 5G7 was captured on the NTA sensor chip at 5 μg/min. A3Bctd-DM at a concentration range of 1 to 500 nM was injected in a running buffer consisting of 10 mM HEPES-NaOH pH 7.5, 150 mM NaCl, 50 μM EDTA, and 0.005%Tween-20 at a flow rate of 30 μL/min. Two independent experiments were performed with the association time of 120 and 300 s and the dissociation time of 300 and 750 s, respectively. The equilibrium dissociation constant (K_D_) was determined by fitting the kinetic data to a 1:1 binding model using Biacore Insight Evaluation software. Biolayer interferometry (BLI) experiments were carried out using the FortéBio Octet Red 384 (Sartorius, Göttingen, Germany). The scFv 5G7 ligand (33 nM) was bound to Ni-NTA biosensors in 10 mM HEPES, 150 mM NaCl, 0.02% Tween-20, 0.1% BSA, pH 7.4. A3 analytes (500, 250, 125, 62.5 nM) were tested for binding with an association time of 600 s and a dissociation time of 900 s, at 30 °C, 1000 rpm.

### 2.4. Immunofluorescence Microscopy

HeLa cells (1 × 10^4^) were transfected with 50 ng of A3Ai-, A3Bi-, A3Gi-mCherry, or empty mCherry vector in a 96-well plate. After 48 h the cells were washed with PBS, fixed in 4% paraformaldehyde for 30 min, washed with PBS, permeabilized in 0.1% Triton-X 100, washed with PBS, and then incubated overnight at 4 °C with anti-A3B 5G7-HA scFv (1:50 dilution) in blocking buffer (5% goat serum, 1% BSA, in PBS). Cells were washed with PBS and incubated in secondary antibody, anti-HA-FITC (1:1000 dilution, Sigma H7411), in blocking buffer for 1 h at room temperature. Cells were washed with PBS and visualized using an Evos-FL microscope (ThermoFisher Scientific).

### 2.5. X-ray Crystallography

Individually purified 5G7 Fv-clasp and A3Bctd-QMΔloop3 proteins were mixed in a 1:1 molar ratio and the complex was isolated by SEC on a 26/600 Superdex 75 column. The isolated complex was concentrated by ultrafiltration to 9.8 mg/mL and subjected to crystallization screening. Crystals were obtained by the sitting drop vapor diffusion method using a reservoir solution consisting of 100 mM Bis-Tris HCl pH 6.5, 400 mM NaCl, and 30% polyethylene glycol 3350. The crystals were cryoprotected with ethylene glycol and flash cooled in liquid nitrogen. X-ray diffraction data were collected at the Northeastern Collaborative Access Team (NE-CAT) beamlines 24-ID-C/E of the Advanced Photon Source (Lemont, IL, USA) and processed using XDS [15]. The structure was determined by molecular replacement phasing with PHASER [16], using the crystal structures of 12CA5 Fv-clasp fragment with its antigen peptide (PDB ID: 5XCS) [17] as the search model. A3Bctd could not be located during molecular replacement, and it was not observed in the final electron density map. Iterative model building and refinement for the 5G7-clasp were performed using COOT [18] and PHENIX [19]. A summary of data collection and model refinement statistics is shown in Table 1. Structure images were generated using PyMOL (https://pymol.org/; accessed on 10 April 2021).

### 2.6. DNA Deamination Assays

A 5′-fluorescein-labeled 15-base DNA oligonucleotide with single TC target sequence for A3B-mediated deamination (5′-TAGGTCATTATTGTG-3′) at 0.2 μM was incubated with 1.0 μM of A3B-DM in the presence of 0 to 20 μM scFv 5G7. After incubation at 37 °C for 70 min, 1.0 μM wild type pfuEndoQ [20] was added and the mixture was further incubated at 60 °C for 30 min. Reactions were stopped by heating to 95 °C for 10 min and adding formamide to 67% by volume. Reaction products were separated on a 15% TBE-urea gel and visualized using a Typhoon FLA 9500 imager.

### 2.7. Computational Docking and MD Simulations

The FTMap computational solvent mapping web server [21] was used in PPI mode to assess potential protein–protein interface regions of A3Bctd. The ClusPro protein–protein docking web server [22] was used to dock the antibody crystal structure obtained in this study (PDB ID: 7KM6) to A3Bctd quadruple mutant with loop3 excised (PDB ID: 5CQI) [8]. Among the 30 poses generated by ClusPro, only 5 poses satisfied the requirement of having an A3Bctd α6 helix at the interface based on mutagenesis studies. All these 5 poses were explored via explicitly solvated, all-atom molecular dynamics (MD) simulations in Amber16 program [23]. Protonation states of titratable amino acids were determined with Schrodinger suite [24]. Each system was explicitly solvated in a TIP3P water box with a buffer distance of 10 Å and ions were added to achieve 0.2 M salt concentration. Amber ff14SB force field parameters [25] were used for protein atoms and Cationic Dummy Atom (CADA) parameters [26] were used for zinc ions and zinc-coordinating residues. Each system was first subjected to multiple steps of energy minimization followed by gradual heating to 310 K and equilibration with decreasing restraints. Following an initial 10 ns MD run to calculate parameters for Gaussian-accelerated MD (GAMD) production runs, each system was simulated in triplicate (1× 275ns + 2× 200ns GAMD) in an NPT ensemble at 310 K with a 2 fs timestep and using dual boost method.

## 3. Results

### 3.1. Biopanning Identifies scFV That Binds Human A3B

A high-affinity binder to a highly soluble A3B C-terminal catalytic domain (ctd) variant [8] was obtained through phage display screening of a naïve scFv library generated from mouse spleen DNA [7]. This scFv was designated 5G7. Library construction and screening procedures were similar to those reported for other targets [9,10]. The original scFv 5G7 obtained from the screen had the light chain Fv domain C-terminus linked via a 20-residue Gly/Ser-rich linker to the N-terminus of the heavy chain Fv domain. We subsequently generated a scFv 5G7 construct with the Fv domains reversed with the heavy chain followed by the light chain and a slightly shortened 16-residue linker containing eight consecutive His residues flanked by tetra-Gly motif (GGGGHHHHHHHHGGGG), which combined to facilitate both expression of soluble protein and purification from *E. coli*.

The purified 5G7 scFv was confirmed to form a stable complex with a nearly wild type A3Bctd construct (residues 187–378 with solubility-enhancing amino acid substitutions L230K and F308K), as judged by co-elution in size-exclusion chromatography analyses (SEC, Figure 1a). Similar results were obtained when 5G7 scFv was co-injected in SEC with A3A, which is ~90% identical in the amino acid sequence to A3Bctd (Figure 1b). In contrast, the 5G7 scFv did not co-elute in SEC with another Z1-type deaminase domain A3Gctd (Figure 1c). A similar selectivity for A3B and A3A, but not A3G, was observed in immunofluorescent microscopy experiments (Figure 1d). In HeLa cells transfected with A3A-cherry and A3B-cherry, the anti-A3B-FITC signal co-localized with Cherry fluorescence (Figure 1d—i,ii). In contrast, the anti-A3B (5G7) scFv did not bind to HeLa cells transfected with A3G-cherry (Figure 1d—iii) or empty mCherry. The binding of scFv 5G7 to A3Bctd was also demonstrated using surface plasmon resonance (SPR), and the equilibrium dissociation constant (K_D_) was determined to be 62 and 79 nM in two independent experiments with different association/dissociation times (Figure 1e). In biolayer interferometry (BLI) experiments, A3B and A3A interacted similarly with 5G7 scFv immobilized on the biosensor, whereas the wavelength shift produced by A3G was much smaller (Figure 1f).

We then tested whether the 5G7 scFv inhibits the DNA deaminase activity of A3Bctd using our recently developed EndoQ-mediated oligonucleotide cleavage assay [20]. The activity of 1.0 μM A3Bctd was not inhibited even in the presence of 20-fold molar excess of the 5G7 scFv (Figure 2), suggesting that this antibody does not bind to or interfere with the active site of A3Bctd. This observation is consistent with the reactivity of the 5G7 scFv to both A3Bctd and A3A as described above, because these two highly homologous proteins differ most significantly in the “loop 1” region flanking the active site and involved in ssDNA-binding [13], in addition to their flexible amino-terminal (interdomain linker for A3B) residues.

### 3.2. Structure Determination of 5G7-Clasp

To facilitate structural studies of the 5G7 scFv, we adapted the recently developed “Fv-clasp” strategy, in which the light and heavy chain Fv domains are held together by an antiparallel coiled coil helices (“SARAH” domain for Sav/Rassf/Hpo) with an additional disulfide bond for further stabilization of the heterodimer [17]. Both “v1” and “v2” versions of 5G7-clasp, which differ in the position of the disulfide linkage, were produced as soluble proteins by co-expression of the light and heavy chains in bacteria. The purified 5G7-clasp of either type (v1 or v2) formed a stable stoichiometric complex with A3Bctd isolable by SEC.

The SEC-purified 5G7-clasp:A3Bctd complexes were subjected to crystallization screening, which yielded crystals of suitable quality for structure determination in multiple conditions. The best crystal diffracted to 1.67 Å resolution and the structure was refined to the free R-factor of 23.4% (Figure 3a–d). Unexpectedly, in all cases, the determined crystal structures showed 5G7-clasp without the bound antigen, A3Bctd. In the crystals, the antigen-binding cleft between the light and heavy chains of 5G7-clasp is occupied by the SARAH domain “clasp” helices, allowing for the formation of an array of 5G7-clasp molecules stacked in a head-to-tail fashion to form the crystal lattice (Figure 3c,d). The SARAH domain also forms lateral contacts in the crystal, forming a bundle of four helices to stabilize self-dimerization of 5G7-clasp. Thus, despite the high stability of the 5G7-clasp-A3Bctd complex, these favorable crystal contacts facilitated by the SARAH helices are likely to have allowed for selective crystallization of free 5G7-clasp. There are two 5G7-clasp molecules in the asymmetric unit of the crystal with essentially identical conformations, with a main chain root mean square deviation of 0.4 Å. Of note, the 5G7-clasp is monomeric (as a covalent heterodimer) in solution, and we did not observe any sign of self-oligomerization, suggesting that the head-to-tail interactions mediated by the clasp helices observed *in crystallo* are not stable in solution.

### 3.3. Protein Interactions in the Antigen-Binding Cleft

In the 5G7-clasp crystal structure, the SARAH domain antiparallel coiled coil α-helices, which serve as the artificial constant region to hold the variable domains together, occupy the antigen-binding cleft of a neighboring molecule *in trans*. The protein–protein interface involves many polar contacts and notably a larger number of tyrosine residues from the antibody, primarily forming hydrogen bonds or other polar interactions, but also making some hydrophobic contacts (Figure 3d). The CDR loops 1 and 2 of the light chain and all CDR loops 1–3 of the heavy chain are involved in direct interaction with the coiled coil α-helices, which run parallel to the light/heavy chain interface and are clamped between the CDR1 loops of the light and heavy chains. All CDR loops are stabilized by an extensive network of hydrogen bonds within each loop as well as with the Fv domain framework regions (Figure 3a,b and Figure 4a–d). CDR3 of the light chain (QQYYS**Y**P**Y**T) is positioned at the bottom of the recess and does not make direct contacts with the bound α-helices. However, a pair of stacked tyrosine residues from this loop (bolded above; Tyr100 and Tyr102) flanking Pro101 in the *cis* peptide bond conformation interact with CDR3 of the heavy chain that features a unique Gly-rich sequence (GGEGYY), which appears to help shape the latter for target binding. The stacked Tyr residues of CDR L3 in turn also stack on a Tyr residue from CDR2 H2, contributing to its unique conformation (Figure 4a,b).

### 3.4. Epitope Mapping by Site-Directed Mutagenesis

A prediction of protein–protein interaction (PPI) surfaces by computational solvent mapping using the FTMap program [21] suggested three distinct areas of A3Bctd surface as possible epitopes for 5G7: the enzyme active site; the region between the N-terminus, β2′ strand, and α2 helix; and that between the C-terminus, α6, and α4 helices (Figure 5). Additionally, we used ClusPro protein–protein docking program [22] to predict possible binding poses between A3Bctd and 5G7. Interestingly, all 30 predicted binding poses pointed to the same epitopes suggested by FTMap analysis except the A3Bctd active site, which was never predicted to be at the binding interface.

Guided by these binding mode predictions and an observation that several Arg residues from the clasp helices play key roles in the interface with 5G7 in the crystal structure above, we performed site-directed mutagenesis to probe the importance of several surface-exposed A3B Arg residues in binding to the 5G7 scFv (Arg252, Arg257, Arg294, Arg304, Arg327, and Arg374). When these amino acid residues were initially mutated in two groups, R252A/R257A showed no effect on the binding of scFv-5G7, as determined by co-elution of the two proteins co-injected in SEC. In contrast, R304E/R327E/R374E completely abolished the complex formation (Figure 6a). We then individually tested the three amino acid substitutions in the latter cluster and, in addition, R294E. The results show clearly that only R374E affects binding, suggesting that Arg374 is essential for the binding of scFv-5G7, whereas the other three Arg residues (Arg294, Arg304, andArg327) do not play significant roles (Figure 6b). A3Bctd R374E also showed no appreciable binding to 5G7 scFv in a BLI experiment (Figure 1f). Taken together, 5G7 is likely to recognize a region including Arg374 within the C-terminal α6 helix of A3Bctd.

### 3.5. Computational Simulations of 5G7-A3Bctd Interaction

Given the availability of high-resolution crystal structures for both A3Bctd and 5G7 antibodies in their free forms and additional experimental restraints on the epitope from site-directed mutagenesis studies, we sought to further investigate possible binding modes by computational simulations. The total of 30 docking poses calculated by ClusPro showed a variety of relative orientations of the two proteins (Figure 7a). In five out of the 30 models, Arg374 of A3Bctd was positioned in the interface with 5G7 (Figure 7b–f). In light of the mutation analysis above, these were considered the most credible candidates. Thus, these five poses were further explored by 3-copy Gaussian-accelerated molecular dynamics (GAMD) simulations (1× 275ns + 2× 200ns), which is an enhanced sampling computational technique [27]. Interestingly, a set of slightly different binding poses (#4 and #22), both of which have A3B Arg374 interacting with 5G7 Asp62 at the C-terminal end of CDR H2, behaved differently in all three copies of MD simulations; pose #4 converged to a stable state, while pose #22 was not stable with A3B Arg374 moving away from 5G7 CDR loops (Figure 7c,f, Appendix A). In another pose (#12), MD simulations converged to a stable binding mode somewhat similar to pose #3 (Figure 7d and Appendix A). The remaining two poses (#3 and #14) have Arg374 of A3Bctd positioned more centered in the antigen-binding cleft between the heavy and light chains (Figure 7b,e). Pose #3 was stable during all three MD copies, whereas significant fluctuations of A3Bctd were observed for #14, particularly high in one of the three MD copies (MD1) (Appendix A).

To probe which of these classes (#4/22 vs. #3/12/14) better represents the actual binding pose, we tested whether a charge-reversing D62R amino acid substitution affects the binding of A3Bctd to 5G7 and whether it rescues the binding defect of A3Bctd R374E. Binding analysis with SEC as above showed that scFv 5G7 D62R bound A3Bctd but failed to bind A3Bctd R374E, not supportive of the importance of the possible ionic interaction between A3B Arg374 and 5G7 Asp62 (Figure 6c). On the other hand, the poses #3 and #14 from the latter class both show the α6 helix of A3B running parallel to the light/heavy chain interface, contacting the CDR3 loops from both chains. Interestingly, the polarity of the α6 helix is completely opposite between these two poses, so that the Arg374 side chain is pointed toward the light chain in one pose (#3), and the heavy chain in the other (#14). Notably, despite their flipped polarity, in either binding pose the enzyme active site of A3Bctd and its DNA-binding groove are not blocked by 5G7, consistent with data described above that scFv 5G7 does not interfere with the DNA deaminase activity of A3B (Figure 2). Considering all MD simulation and mutagenesis data including stability during the simulations (Appendix A), we conclude that pose #3 is the most likely binding mode of 5G7 to A3B.

## 4. Discussion

The in vitro selected minimal antibody 5G7 binds A3Bctd with a reasonable affinity for a scFv selected from a naïve library (K_D_ ~70 nM). While the 5G7 scFv does not apparently discriminate between A3A and A3Bctd, it does show selectivity against A3Gctd (the only other Z1-type deaminase domain in humans). Although each of these three proteins features a unique active site environment underlying their distinct enzymatic properties including different target DNA selectivity [13,28,29,30], they share significant overall amino acid sequence homology. In particular, A3Bctd residues in the C-terminal α6 helix (^368^ALSGRLRAILQ^378^) flanking Arg374, which was identified as key in the A3Bctd-5G7 interaction in our studies, are highly conserved between A3A, A3Bctd, and A3Gctd (Figure 5). Thus, it is not surprising that 5G7 scFv has affinity toward A3A. Interestingly, residues 354–382 spanning the α6 helix were used as the peptide immunogen in raising the recently reported anti-A3B rabbit monoclonal antibody 5210-87-13, for which cross-reaction with A3A and A3G is clearly documented [6]. It is possible that the 5G7 scFv independently derived from in vitro selection shares a similar mode of A3Bctd binding with 5210-87-13. However, we did not observe stable complex formation between a peptide fragment corresponding to the α6 helix of A3Bctd with the 5G7 scFv, suggesting that the binding mechanisms between 5G7 and 5210-87-13 are not exactly the same. In this regard, it would be interesting to further investigate whether the selectivity of 5G7 toward A3Bctd over A3Gctd is based on their difference within the α6 helix (Ala368 of A3B vs. Asp370 of A3G) or outside the α6 helix (Appendix A).

Computational simulations and site-directed mutational analyses combined to provide constraints on plausible binding poses for the A3Bctd–5G7 interaction, which suggested that the C-terminal α6 helix of A3Bctd may fit in the cleft between the light and heavy chains of 5G7. Residual ambiguity, including the polarity of α6 helix, could be resolved with additional experimental restraints. Notably, the positioning of A3Bctd α6 helix mimics the crystal packing interaction observed in the 5G7-clasp crystal structure, where two antiparallel “clasp” helices occupy the antigen-binding cleft of the 5G7 antibody. Favorable binding of positively charged α-helical surfaces in the antigen-binding cleft of 5G7 might have promoted the formation of the 5G7_clasp crystal lattice, despite our attempts to crystallize the A3Bctd-5G7 complex. Nonetheless, our high-resolution structure of the 5G7 antibody in combination with their computational and biochemical characterization may help engineer A3B antibodies with higher affinity and selectivity useful in immunological applications in studying virus–host interactions and cancer biology.

## Figures and Tables

**Figure 1 viruses-13-00663-f001:**
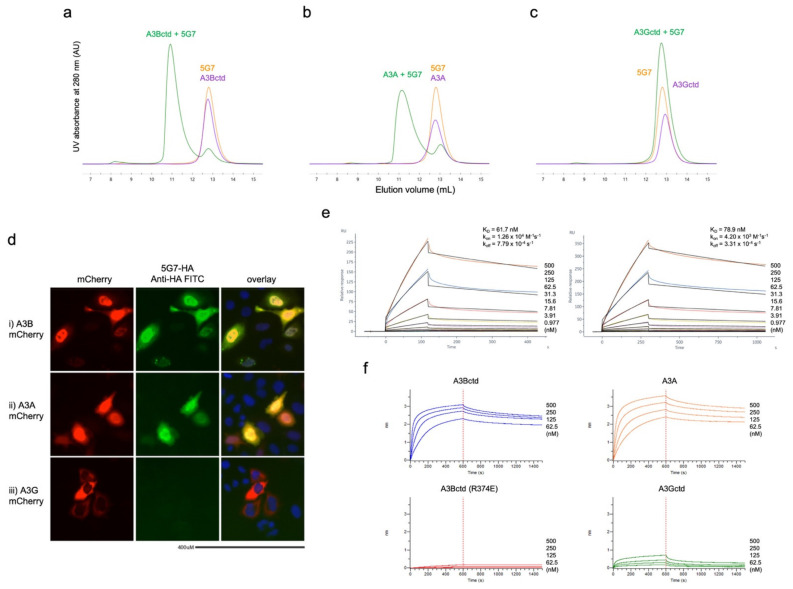
**scFv 5G7 binds A3Bctd and A3A but not A3Gctd.** (**a**–**c**) Size-exclusion chromatography profiles showing the co-elution of A3Bctd (**a**) and A3A (**b**) with scFv 5G7 as stable binary complexes. In contrast, A3Gctd (**c**) and scFv 5G7 eluted as independent peaks (overlapping due to their similar sizes). (**d**) Immunofluorescence to detect A3B in HeLa cells transiently transfected with (**i**) A3Ai-Cherry, (**ii**) A3Bi-Cherry, or (**iii**) A3Gi-Cherry. (**e**) SPR sensorgrams showing the interaction of A3Bctd with immobilized scFv 5G7. Two independent experiments were performed with the association time of 120 and 300 s and the dissociation time of 300 and 750 s, respectively. The equilibrium dissociation constant (K_D_) was determined by fitting the kinetic data to a 1:1 binding model (colored lines show the best fit). (**f**) Biolayer interferometry (BLI) sensorgrams showing the interaction of A3Bctd, A3A, A3Gctd, and A3Bctd (R374E) to immobilized scFv 5G7. Because the observed kinetics did not follow a 1:1 binding model very well, K_D_ values were not determined.

**Figure 2 viruses-13-00663-f002:**
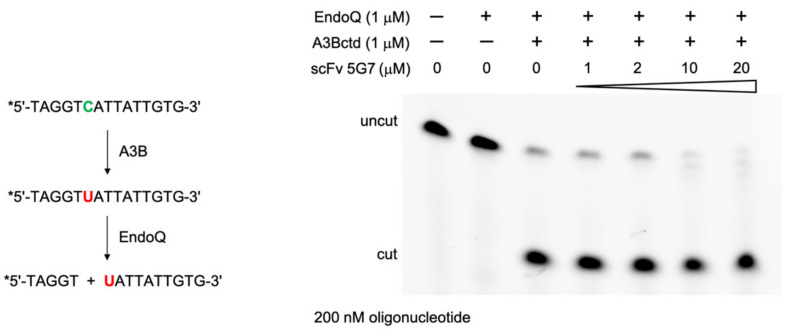
**scFv 5G7 does not inhibit A3B DNA deaminase activity.** Schematic diagram (left) shows the principle of the DNA cytosine deaminase assay mediated by EndoQ [20]. Fluorescent label on the 5’ end of the oligo DNA substrate is denoted by an asterisk (*). The assay result (gel image on the right) shows that the addition of scFv 5G7 up to 20 μM had no inhibitory effect on the A3B activity. At higher concentrations of scFv 5G7, the residual uncut DNA band shows a ladder pattern, which is probably due to a contaminating exonuclease activity in the scFv 5G7 preparation.

**Figure 3 viruses-13-00663-f003:**
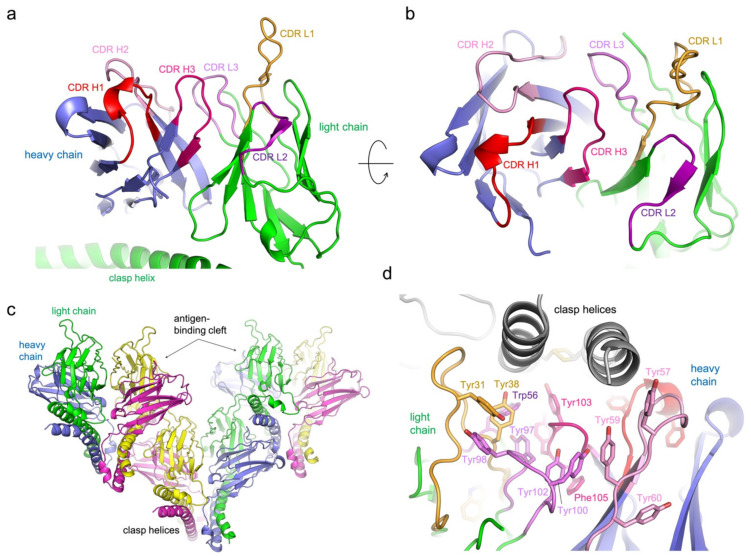
**Crystal structure of the 5G7-clasp.** (**a**) An overall view of 5G7, with the CDR loops 1–3 of each (heavy and light) chain colored differently. (**b**) Close-up view of the antigen-binding cleft. (**c**) Observed crystal packing showing head-to-tail stacking of 5G7-clasp. (**d**) Binding of the clasp helices in the antigen-binding cleft. The abundance of aromatic side chains of 5G7 is highlighted.

**Figure 4 viruses-13-00663-f004:**
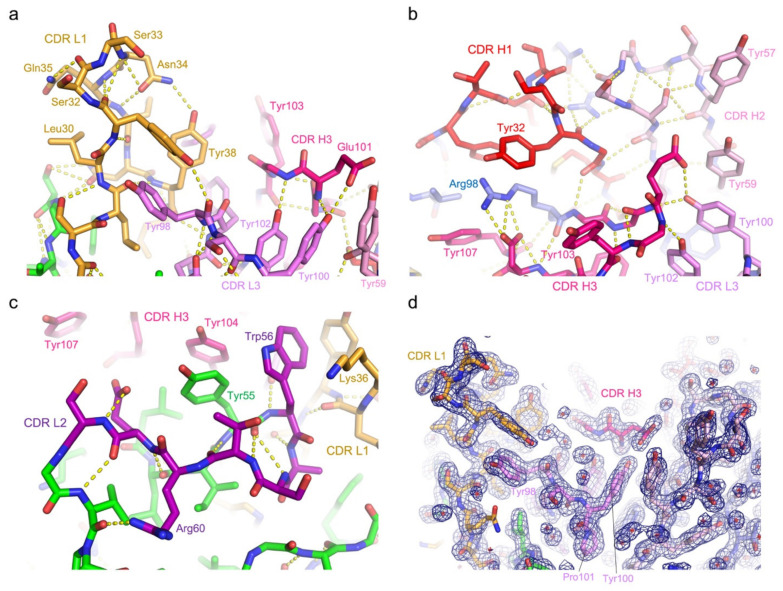
**Network of hydrogen bonds stabilizing the CDR loops.** (**a**–**c**) Close-up views of three different regions showing the extensive network of hydrogen bonds to stabilize the CDR loops. (**d**) 2mFo-DFc electron density map contoured at 1.0 σ overlaid on the atomic model.

**Figure 5 viruses-13-00663-f005:**
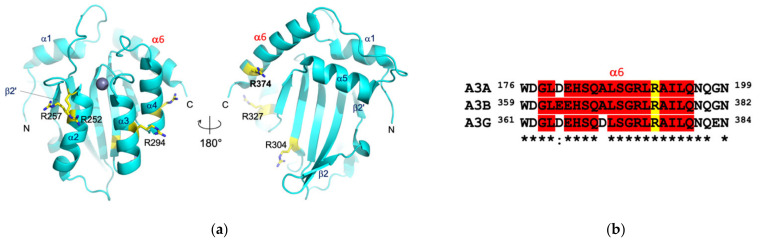
**A3Bctd structure and the C-terminal α6 sequences of Z1-type deaminases.** (**a**) Crystal structure of A3Bctd [8], with the locations of the surface-exposed Arg residues mutated in this study and the C-terminal α6 helix highlighted. (**b**) Alignment of α6 sequences from A3A, A3Bctd, and A3Gctd. The conserved Arg residue (Arg374 in A3B) is in yellow.

**Figure 6 viruses-13-00663-f006:**
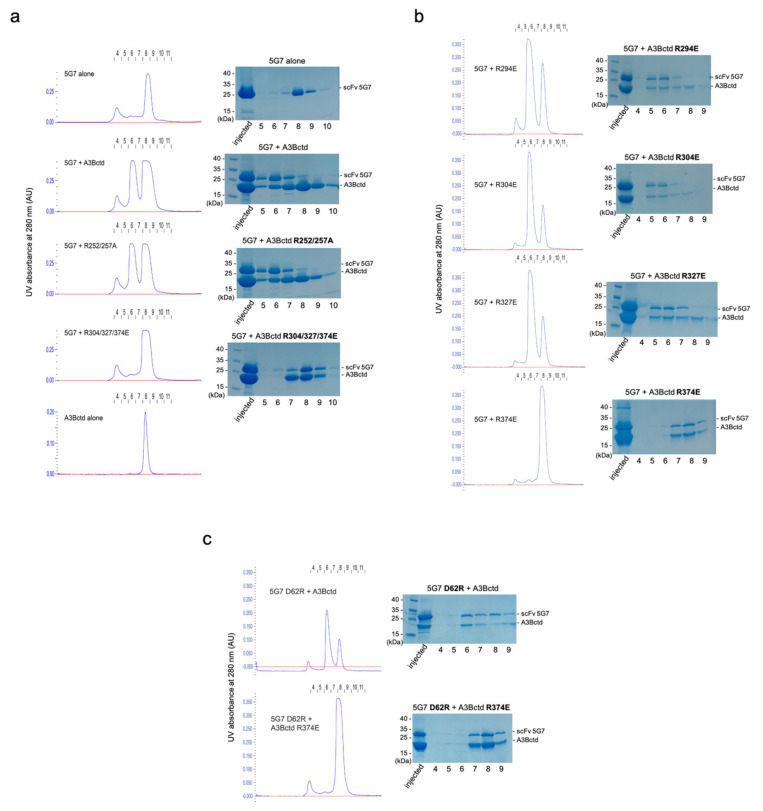
**A3Bctd R374 is essential in 5G7-binding.** (**a**) Complex formation by A3Bctd variants with substitutions of multiple Arg residues. The chromatograms are shown on the left, and SDS-PAGE analyses of the collected fractions are shown on the right. (**b**) Complex formation by A3Bctd variants with individual Arg substitutions. The results show that A3Bctd Arg374 is required for the binding to 5G7 while other Arg residues are not. (**c**) Complex formation between 5G7 D62R and A3Bctd or A3Bctd-R374E, showing that Asp62 is not important in the binding of A3Bctd.

**Figure 7 viruses-13-00663-f007:**
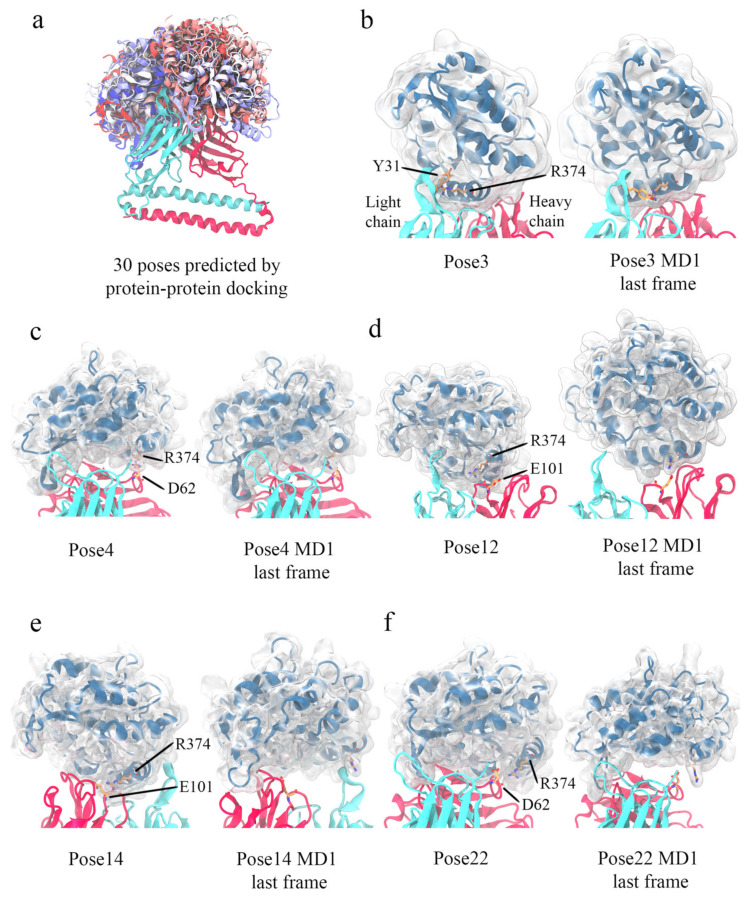
**Possible A3Bctd-5G7 binding poses.** (**a**) All 30 binding modes of A3B-5G7 predicted by ClusPro program. 5G7 is depicted in cyan and red ribbons for light and heavy chains, respectively. Each A3B pose is shown in ribbons colored differently in a spectrum from blue to red. (**b**–**f**) The five binding modes that have A3B Arg374 at the interface. For each binding pose, the initial binding mode is shown side-by-side with the final frame of MD1. 5G7 is depicted the same as in (**a**). A3B is shown with blue ribbons in addition to its transparent protein surface. For each case, A3B Arg374 and its interacting partner in 5G7 are depicted in sticks with C atoms in orange, O atoms in red, and N atoms in blue.

**Table 1 viruses-13-00663-t001:** Summary of X-ray data collection and model refinement statistics.

	5G7-Clasp
Data Collection	
Wavelength (Å)	0.979
Resolution range (Å)	84.9–1.67 (1.73–1.67)
Space group	P2_1_
Unit cell	
* a, b, c* (Å)	54.07 68.82 88.76
α, β, γ (°)	90 107.05 90
Total reflections	269,642 (25,086)
Unique reflections	71,734 (7125)
Multiplicity	3.8 (3.5)
Completeness (%)	98.54 (98.81)
*<I*/*σ(I)>*	7.88 (1.09)
*R* _merge_	0.1088 (0.8745)
*R* _meas_	0.1267 (1.021)
*R*_p_._i_._m_.	0.06402 (0.518)
CC_1/2_	0.997 (0.879)
Refinement	
Reflections for R_work_	71,268 (7117)
Reflections for R_free_	3330 (333)
*R* _work_	0.192 (0.362)
*R* _free_	0.234 (0.367)
No. of non-H atoms	5935
Macromolecules	5489
Ligands	29
Solvent	417
Protein residues	672
R.m.s. deviations	
Bond length (Å)	0.006
Bond angles (°)	0.85
Ramachandran plot	
Favored (%)	98.49
Allowed (%)	1.51
Outliers (%)	0.00
Average B-factor	32.30
Macromolecules	31.46
Ligands	52.16
Solvent	41.95

Statistics for the highest-resolution shell are shown in parentheses.

## Data Availability

The atomic coordinates and structure factors for 5G7 Fv-clasp have been deposited in the RCSB Protein Data Bank, with the accession code 7KM6.

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
