# Peer review of "Structural Characterization of a Minimal Antibody against Human APOBEC3B"

_viruses, 2021, doi:10.3390/v13040663_

Round 1

Reviewer 1 Report

In this manuscript “Structural characterization of a minimal antibody against human APOBEC3B” by Tang et al., the authors described single chain variable fragment (scFv) that specifically binds to both A3B and A3A, but not A3G. They obtained the scFV through phage display screening, and characterized it by immune-assays, and determined its x-ray crystallography to very high resolution (1.8A). From table statistics, it appears the resolution of the crystal form can be higher than the reported resolution, if the authors want to push it, as the high-resolution bin statistics shows very strong reflection. They also performed simulations of how A3B-scFv might bind scFv and used site-directed mutagenesis to identify a key residue involved in the A3B-scFv interaction. It would be helpful to show or discuss if this antibody has any cross-reaction with any other APOBEC members. While this is not directly related to the studies of viruses pe se, it is about the characterization for a reagent that can be used to study A3B and A3A, both of which have a role in innate immunity. As such, I think it’s suitable for publication in this journal.

Author Response

In this manuscript “Structural characterization of a minimal antibody against human APOBEC3B” by Tang et al., the authors described single chain variable fragment (scFv) that specifically binds to both A3B and A3A, but not A3G. They obtained the scFV through phage display screening, and characterized it by immune-assays, and determined its x-ray crystallography to very high resolution (1.8A). From table statistics, it appears the resolution of the crystal form can be higher than the reported resolution, if the authors want to push it, as the high-resolution bin statistics shows very strong reflection. They also performed simulations of how A3B-scFv might bind scFv and used site-directed mutagenesis to identify a key residue involved in the A3B-scFv interaction. It would be helpful to show or discuss if this antibody has any cross-reaction with any other APOBEC members. While this is not directly related to the studies of viruses pe se, it is about the characterization for a reagent that can be used to study A3B and A3A, both of which have a role in innate immunity. As such, I think it’s suitable for publication in this journal.

We appreciate this helpful suggestion. We agree that the diffraction data are useful beyond our initial resolution cutoff (1.88 Å) and have re-refined the structure to 1.67 Å. The x-ray data and model refinement statistics in Table 1 have been updated accordingly. The electron density figure has also been updated (Fig. 4d), which shows noticeable improvement. We have not tested if 5G7 recognizes the Z2 or Z3-type deaminase A3C/D/F/H but given that they are more distantly related to A3Bctd than A3Gctd, we strongly suspect that they do not cross-react.

Reviewer 2 Report

This manuscript describes new results from the continuing efforts of the Harris and Aihara labs to create new reagents and molecular structures to aid research on human APOBEC3 members. They have previously described monoclonal antibodies raised against C-terminal peptide from APOBEC3B (A3B) and here they describe a single-chain variable fragment (scFV) called 5G7 obtained through phage display biopanning and characterize it in terms of binding to A3B carboxyterminal domain (A3Bctd), APOBEC3A (A3A) or Apobec3G carboxyterminal domain (A3Gctd) and the effect of 5G7 on the catalytic activity of A3Bctd. This is useful information for investigators who may want to use 5G7. However, the remaining manuscript involves structural studies that are less useful and where the results are less conclusive. First, an attempt to generate a structure of a modified form of 5G7 with A3Bctd gives rise to only the structure of 5G7 and the structure shows that the modifications they made to the 5G7 may have prevented its binding to A3Bctd. Using computational analysis of binding sites and site-directed mutagenesis they conclude that only Arg374 is essential for the binding of 5G7 to A3Bctd. This is despite the fact that Arg374 and its flanking residues are completely conserved in A3Gctd, which does not bind 5G7. Furthermore, they note that they did not find a stable complex between 5G7 and the α helix polypeptide that contains  Arg374, further muddying the issue. They also perform MD simulations that suggest a possible mode of interaction between 5G7 and A3Bctd, but the usefulness of this analysis is unclear. Overall, this antibody fragment 5G7 will be useful in distinguishing A3A/A3B from A3G, but the structural analysis presented here is of dubious value. It could be eliminated from the manuscript altogether. The following experiments would improve the usefulness of 5G7 protein for researches in the virology and cancer biology fields-

  1. Use full-length A3B- not A3B-CTD- for binding and inhibition studies.
  2. Perform binding studies also with A3A. It would be useful to know the relative affinities of 5G7 to A3B vs. A3A.
  3. Determine whether or not 5G7 may be used in Western blots and pull-down assays.

Other minor comments:

  1. The text in Figure 6 is unreadable.
  2. Label the antigen-binding cleft and the "clasp" helices in Fig. 3C.
  3. Figure 2- The A3Bctd activity appears to go up (no substrate left) with increasing amounts of 5G7. Is this real or is it due to contaminating nuclease activity?
  4. Fig. 1d- It would be better to stack all four sets of panels A3A, A3B, A3G and mCherry alone vertically. In the current arrangement, moving horizontally, A3A images change to A3B images and A3G images change to mCherry images. This is confusing.
  5. The manuscript uses terminology such as "poses", SARAH domain and 3-copy Gaussian-accelerated molecular dynamics (GAMD) simulations (1x 275ns + 2x 200ns) that are likely to be unfamiliar to most readers. Some explanation would be useful.
  6. The manuscript repeatedly points out that A3B has a Z1-type Zn-binding domain. As the interaction between 5G7 and A3Bctd does not involve Zn2+ and its neighbors, the reasons for pointing this out are unclear.

Author Response

This manuscript describes new results from the continuing efforts of the Harris and Aihara labs to create new reagents and molecular structures to aid research on human APOBEC3 members. They have previously described monoclonal antibodies raised against C-terminal peptide from APOBEC3B (A3B) and here they describe a single-chain variable fragment (scFV) called 5G7 obtained through phage display biopanning and characterize it in terms of binding to A3B carboxyterminal domain (A3Bctd), APOBEC3A (A3A) or Apobec3G carboxyterminal domain (A3Gctd) and the effect of 5G7 on the catalytic activity of A3Bctd. This is useful information for investigators who may want to use 5G7. However, the remaining manuscript involves structural studies that are less useful and where the results are less conclusive. First, an attempt to generate a structure of a modified form of 5G7 with A3Bctd gives rise to only the structure of 5G7 and the structure shows that the modifications they made to the 5G7 may have prevented its binding to A3Bctd. Using computational analysis of binding sites and site-directed mutagenesis they conclude that only Arg374 is essential for the binding of 5G7 to A3Bctd. This is despite the fact that Arg374 and its flanking residues are completely conserved in A3Gctd, which does not bind 5G7. Furthermore, they note that they did not find a stable complex between 5G7 and the α helix polypeptide that contains  Arg374, further muddying the issue. They also perform MD simulations that suggest a possible mode of interaction between 5G7 and A3Bctd, but the usefulness of this analysis is unclear. Overall, this antibody fragment 5G7 will be useful in distinguishing A3A/A3B from A3G, but the structural analysis presented here is of dubious value. It could be eliminated from the manuscript altogether. The following experiments would improve the usefulness of 5G7 protein for researches in the virology and cancer biology fields-

We thank the reviewer for helpful comments.

1. Use full-length A3B- not A3B-CTD- for binding and inhibition studies.

Full-length A3B has solubility issues, which prevent many biochemical studies (i.e. essentially any experiment requiring highly concentrated, pure protein). We therefore focused on A3Bctd in our present studies, which is also where the epitope/binding site for 5G7 is located.  

However, to help address this reviewer’s point, we have conducted immunofluorescent microscopy experiments and now show that the 5G7 antibody binds to full-length A3B in living cells.

2. Perform binding studies also with A3A. It would be useful to know the relative affinities of 5G7 to A3B vs. A3A.

We have added biolayer interferometry (BLI) data as a new figure panel (Fig. 1f), which show in parallel the binding of scFv 5G7 to A3Bctd, A3A, A3Gctd, and A3Bctd(R374E). The BLI sensorgrams suggest that A3Bctd and A3A interact with 5G7 scFv with a similar affinity. Please note that because of somewhat complex association and dissociation kinetics, we chose not to fit these data to a 1:1 binding model to derive KD values.

3. Determine whether or not 5G7 may be used in Western blots and pull-down assays.

5G7 does not react with denatured A3Bctd and thus it did not work in Western blots. We suspect that pull-down assays in native conditions may work, especially given the specific and strong staining in immunofluorescence experiment shown in updated Fig. 1d.

Other minor comments:

1. The text in Figure 6 is unreadable.

The small font sizes have been corrected.

2. Label the antigen-binding cleft and the "clasp" helices in Fig. 3C.

We labeled the antigen-binding cleft and the clasp helices as suggested.

3. Figure 2- The A3Bctd activity appears to go up (no substrate left) with increasing amounts of 5G7. Is this real or is it due to contaminating nuclease activity?

We believe this was due to a residual nuclease activity in the scFv prep.

4. Fig. 1d- It would be better to stack all four sets of panels A3A, A3B, A3G and mCherry alone vertically. In the current arrangement, moving horizontally, A3A images change to A3B images and A3G images change to mCherry images. This is confusing.

We updated the figure as suggested.

5. The manuscript uses terminology such as "poses", SARAH domain and 3-copy Gaussian-accelerated molecular dynamics (GAMD) simulations (1x 275ns + 2x 200ns) that are likely to be unfamiliar to most readers. Some explanation would be useful.

We introduced the SARAH domain acronym by spelling it out (Sav/Rassf/Hpo domain) on its first appearance. We also added a brief explanation and a reference for GAMD (Ref #27).

6. The manuscript repeatedly points out that A3B has a Z1-type Zn-binding domain. As the interaction between 5G7 and A3Bctd does not involve Zn2+ and its neighbors, the reasons for pointing this out are unclear.

We use the term ‘Z1-type deaminase domain’ in several places, which is an established nomenclature for the class of cytidine deaminases including A3Bctd.

Reviewer 3 Report

Editing deaminases of the A3 family participate in antiviral responses and contribute to mutagenesis in cancer. Generation of reagents that help understanding A3 biology is a critical mission.  The current manuscript describes the structure of minimal antibody against A3B and the computational model of the putative complex structure. Unfortunately, the antibody-A3B complex crystals possessed only the complex's antibody part, raising concerns about the structure's biological relevance. To address that, the authors did epitope mapping by site-directed mutagenesis, which, in combination with computer modeling, provided the most likely binding mode of antibody to A3B. The work may advance the technology of the creation of antibodies against A3s. The authors do not go beyond that technical aspect. They write that the generation of antibodies against individual family members is desired but do not elaborate on possible practical applications of such antibodies.   

Abstract: needs more clarity.

Line 25. What is meant by “A5B …as a component of multiple human cancers”

26. Explain why A3B antibodies are desirable.

38. Clarify “….these antibodies from very different sources”

Introduction: is the significance of the study is just a mere technical advance?

47. Sounds strange “7 enzymes in most humans…”

51. Delete “phenotypes”?

452-6. The sentence looks clumsy.

45. Word combination "basic research" is to generic to convince the readers of the study's significance.

Materials and Methods: some polishing would be good.

96-98. The sentence reads as ” previously crystallized …were expressed”. Maybe the authors could find a way to omit the duplication of the word “previously”?

Results.

180. Binder to “crystallized” or a protein whose structure is known?

208. Can the authors comment on the effect of apparent deamination stimulation by the antibody?

387-8. Explain the axes units. Image of representative PAGE gel would be good to present, like in Fig. 6.

Discussion: can the authors explain why antibody does not bind to A3G by similar modeling? This simulation could strengthen the validity of the modeling.

Author Response

Editing deaminases of the A3 family participate in antiviral responses and contribute to mutagenesis in cancer. Generation of reagents that help understanding A3 biology is a critical mission.  The current manuscript describes the structure of minimal antibody against A3B and the computational model of the putative complex structure. Unfortunately, the antibody-A3B complex crystals possessed only the complex's antibody part, raising concerns about the structure's biological relevance. To address that, the authors did epitope mapping by site-directed mutagenesis, which, in combination with computer modeling, provided the most likely binding mode of antibody to A3B. The work may advance the technology of the creation of antibodies against A3s. The authors do not go beyond that technical aspect. They write that the generation of antibodies against individual family members is desired but do not elaborate on possible practical applications of such antibodies.   

We thank the reviewer for helpful comments.

Abstract: needs more clarity.

Line 25. What is meant by “A5B …as a component of multiple human cancers”

For clarity, “component” was changed to “driver”.

  1. Explain why A3B antibodies are desirable.

We added “for its specific detection in various research and possibly diagnostic applications”.

  1. Clarify “….these antibodies from very different sources”

The phrase “these antibodies from very different origins” refers to the rabbit monoclonal antibody, mAb 5210-87-13, and 5G7 obtained in a phage display screening of a mouse-spleen DNA-derived scFv library. We would like to keep it concise as this sentence is in Abstract.

Introduction: is the significance of the study is just a mere technical advance?

  1. Sounds strange “7 enzymes in most humans…”

The A3 family consists of 7 members (A3A/B/C/D/F/G/H), but there is a common deletion polymorphism that removes the APOBEC3B gene in some human population.

  1. Delete “phenotypes”?

The phrase “clear subcellular localization phenotypes” has been changed to “distinct subcellular localization patterns”

452-6. The sentence looks clumsy.

We were unable to find the sentence at the indicated line number.

  1. Word combination "basic research" is to generic to convince the readers of the study's significance.

To be more specific, we changed the phrase to “studies on A3 expression, localization, and interaction partners”.

Materials and Methods: some polishing would be good.

96-98. The sentence reads as ” previously crystallized …were expressed”. Maybe the authors could find a way to omit the duplication of the word “previously”?

The second “previously” was deleted.

Results.

  1. Binder to “crystallized” or a protein whose structure is known?

The confusing sentence was fixed by removing “previously crystallized”.

  1. Can the authors comment on the effect of apparent deamination stimulation by the antibody?

We believe that degradation of the residual substrate DNA at higher antibody concentrations was due to a contaminating nuclease activity in the scFv prep.

387-8. Explain the axes units. Image of representative PAGE gel would be good to present, like in Fig. 6.

Axes labels have been added in Fig. 1a-c. We agree that PAGE gel would have been good but unfortunately, we do not have gel images from these particular experiments.

Discussion: can the authors explain why antibody does not bind to A3G by similar modeling? This simulation could strengthen the validity of the modeling.

A possible explanation is that Asp370 of A3G (corresponding to Ala368 of A3B) in a6 helix interferes with 5G7 binding. This is now mentioned in Discussion and highlighted in a newly added supplementary Fig. S6.